# Exploring nomophobia with a German adaption of the nomophobia questionnaire (NMP-Q-D)

**Melina Coenen, Yvonne Görlich**[ID]*

PFH Private University of Applied Science Göttingen, Göttingen, Germany

* goerlich@pfh.de

**Data Availability Statement:** All relevant data are within the manuscript and its Supporting information files.

**Funding:** The authors received no specific funding for this work.

## Abstract

Nomophobia is considered a disorder of the modern world and describes the fear of being separated from one's smartphone and being no longer connected and reachable. The aims of this Study were to translate the nomophobia questionnaire (NMP-Q) into German, validate the NMP-Q-D, and use it to shed light on the nomophobia construct. A total of 807 volunteer test subjects were included in the evaluation, 50 of them participated five months later in a retest study. A 4-factor structure of the NMP-Q-D could be confirmed by exploratory as well as by confirmatory factor analyses. The four factors are: (1) "Not being able to communicate", (2) "Losing connectedness", (3) "Not being able to access information", and (4) "Giving up convenience". The Cronbach's alpha coefficient of the NMP-Q-D was .92 and the test-retest-reliability was .80. Significant correlations of frequency of smartphone usage with time spent confirmed criterion validity of NMP-Q-D. Construct validity was given by significant correlations of NMP-Q-D to fear of missing out and smartphone addiction. Neuroticism was positively associated with nomophobia, while consciousness and openness were lightly negatively associated. Anxiety correlated significantly positively with factor 1, and stress with factors 1 and 4. Life satisfaction was positively associated with factor 3 and well-being negatively with factor 4. A multiple regression analysis revealed smartphone usage, gender, and neuroticism as significant predictors of nomophobia. Females scored significantly higher for factors 1 and 4 compared to males. Nomophobia was rather widespread in the sample: Nearly half of the participants (49.4%) had a moderate level of nomophobia and 4.1% a severe nomophobia.

## Introduction

Smartphones are an integral part of everyday life. In 2021, there were 6.259 billion smartphone subscriptions worldwide [1]. A total of 62.61 million people in Germany use smartphones, which corresponds to around 77.9% of the German population [2]. This puts Germany in third place worldwide [3] after the United Kingdom (78.9%) and the United States (81.6%). In Germany, the average smartphone usage is about 229 minutes (3 hours and 49 minutes) per day [4].

**Competing interests:** The authors have declared that no competing interests exist.

Smartphone usage results in many positive consequences, such as the ability to communicate and stay in touch with friends and family during the Covid-19 pandemic. On the other hand, excessive smartphone usage can lead to various negative psychological consequences, such as nomophobia [5]. Nomophobia stands for "no mobile phone phobia" and is considered a disorder of the modern world [6]. Nomophobia occurs primarily due to excessive smartphone usage and describes the fear of being disconnected and unavailable without one's smartphone. This can happen especially when the battery is empty, there is no network reception, or the smartphone has been forgotten [5–7]. Despite much overlap with smartphone and internet addiction, nomophobia represents a separate construct. Unlike nomophobia, smartphone addiction is not a pathological anxiety. Instead, sufferers primarily experience a loss of control over their smartphone usage which affects other areas of their lives [7–9]. Another phenomenon closely related to nomophobia is the fear of missing out (FoMO). A strong correlation between nomophobia and FoMO could be shown in a previous study [10].

Furthermore, significant associations were found between nomophobia and loneliness, depression, distraction, and decreased impulse control [6]. 83% of students reported that they had already experienced panic attacks because they had misplaced their smartphones. Side effects such as headaches and sluggishness were also shared by 61% of the students [11]. A systematic review shows significantly positive correlations between depression, stress, and anxiety with problematic smartphone usage [12]. Meta-analytically, an effect size of $r = .22$ was found between smartphone usage, stress and anxiety [13].

Yildirim and Correia developed the NMP-Q Nomophobia-Questionnaire to measure nomophobia [7]. It contains 20 items and was validated on a sample of 301 students. The items can be grouped into four factors, which explained a total of 69.6% of the variance. These four factors comprise: (1) "not being able to communicate", (2) "losing connectedness", (3) "not being able to access information", and (4) "giving up convenience". The response scale of the NMP-Q is a seven-point Likert scale with 1 = "Strongly disagree" to 7 = "Strongly agree". Scores can range from 20 to 140, with higher scores corresponding to a stronger expression of nomophobia. A score of 20 indicates no nomophobia, scores between 21 and 59 correspond to mild, 60 and 99 to moderate, and 100 or higher to severe nomophobia. The NMP-Q has good psychometric characteristics with a Cronbach's alpha of .95 and a construct validity of $r = .71$ [7] as a correlation to the Mobile Phone Involvement Questionnaire (MPIQ) [14]. The NMP-Q has been translated into other languages, including Spanish, Italian, Portuguese, and Chinese [9, 15–17].

So far, there was no German nomophobia questionnaire. Therefore, one aim of this study was to translate and validate the NMP-Q.

## Materials and methods

### Procedure

First, we sought permission from the authors of the NMP-Q to translate the questionnaire and received a confirmation from Dr. Ana-Paula Correia. The translation of the NMP-Q into German took place in several steps and was oriented towards both the translation-back-translation principle and the "TRAPD" procedure [18]. In the first step, two independent translations of the questionnaire were made by the two authors of this study. Subsequently, they discussed the results and agreed on one translation. The focus was primarily on a translation that was correct in meaning. In the next step, a back translation of the 20 items was prepared by a computer scientist (native German speaker, fluent in English). This back translation was very similar to the original; however, as expected, it did not completely match the original, as the focus was on an understandable and meaningful translation rather than a word-by-word translation. For this

reason, an additional content review of the translation was carried out by an engineer (native German and English speaker). The focus was on whether the German translation corresponds to the English original. Based on the feedback, a small correction was made to the content of item 8. Otherwise, it was confirmed that all other items of the German translation matched the original English version. Finally, the translated questionnaire (NMP-Q-D) was presented to a group of six people to check the comprehensibility of the items. No incomprehensibility was reported. The final German translation of the NMP-Q items is listed in Table 1.

The subsequent implementation of the study took place online in a cross-sectional design. In May 2021, the first data collection was performed through the LimeSurvey platform. The subjects were recruited using the snowball principle. This included sending the survey to various WhatsApp groups. Additionally, the survey was shared via the social media platform "Instagram". Thus, the sample is a convenience sample. Requirements for participation in the survey were a minimum age of 18 years and ownership of a smartphone. Subjects did not receive any compensation. A repeat measurement was performed in October 2021, again through the LimeSurvey platform.

## Participants

A total of 807 subjects were included in the study, 721 females (89.3%), 75 males (9.3%), 3 diverse (0.4%) and 2 undefined (0.2%). The average age was 25 years (SD = 9.26; range 18 to 87). 97 (12.0%) persons were students at school, 95 (11.8%) students at vocational school, 290 (35.9%) students at university, 339 (42.0%) were working, 27 (3.3%) persons were on parental leave or housewives/ househusbands, 12 (1.5%) were retired, and 20 (2.5%) unemployed. Regarding the level of education, most of the test persons (583, 72.2%) had a high school degree, 192 (23.8%) a middle school leaving certificate, 23 (2.9%) a lower secondary school certificate, and 6 (0.7%) had not graduated from school at the time of completing the survey.

A repeat measurement was carried out 5 months later (M = 5.02 months; SD = 0.14 months) with 50 test persons (45 females (90%), 5 males (10%)), age: M = 24.10 years, SD = 6.12, range 18 to 56), 2 (4%) persons were students at school, 3 (6%) at vocational school, 26 (52%) at university, 16 (32%) were working, 5 (10%) on parental leave or housewives/ househusbands, and 1 (2%) unemployed; 42 (84%) had a higher education degree and 8 (16%) a middle school leaving certificate.

## Measurements

In addition to the NMP-Q-D, the online surveys also included questions on demographic data and smartphone usage at both measurement times. For smartphone usage, the subjects were asked how frequently (activation) and for how long (in hours and minutes) they use their smartphone per day. Furthermore, at measurement time 1, the following questionnaires were used for validation purposes:

The Fear of Missing Out Scale (FoMO) contains ten items and a five-point Likert scale (from 1 = not at all true of me to 5 = extremely true of me). The FoMO score is computed by averaging responses to all ten items [19, 20]. Cronbach's alpha was .87 [19], and in the present study it was .77.

The short version of the Smartphone Addiction Scale (SAS-SV) measures smartphone addiction. It contains ten items, with a six-point Likert scale (from 1 = Strongly disagree to 6 = Strongly agree). The sum of these items gives an overall SAS-SV score (range: 10–60) [21]. Overall, the German version of the SAS-SV shows good psychometric properties with an internal consistency of $\alpha$ = .85 [22]. In our study, Cronbach's alpha was .83.

**Table 1. Items of the German version of the nomophobia questionnaire (NMP-Q-D).**

| English original Items in the NMP-Q ([7], Table 1, p. 134) | German translation |
|---|---|
| 1. I would feel uncomfortable without constant access to information through my smartphone | 1. Ich würde mich unwohl fühlen, wenn ich keinen ständigen Informationszugang durch mein Smartphone hätte. |
| 2. I would be annoyed if I could not look information up on my smartphone when I wanted to do so | 2. Es würde mich ärgern, wenn ich keine Informationen auf meinem Smartphone abrufen könnte, wann immer ich möchte. |
| 3. Being unable to get the news (e.g., happenings, weather, etc.) on my smartphone would make me nervous | 3. Es würde mich nervös machen, wenn ich keine Nachrichten (z.B. Ereignisse, Wettervorhersagen etc.) über mein Smartphone abrufen könnte. |
| 4. I would be annoyed if I could not use my smartphone and/or its capabilities when I wanted to do so | 4. 4. Es würde mich ärgern, wenn ich mein Smartphone und/oder dessen Funktionen nicht benutzen könnte, wann immer ich möchte. |
| 5. Running out of battery in my smartphone would scare me | 5. Ein leerer Akku in meinem Smartphone würde mir Angst machen. |
| 6. If I were to run out of credits or hit my monthly data limit, I would panic | 6. Wenn ich kein Guthaben mehr hätte oder mein monatliches Datenvolumen aufgebraucht wäre, würde ich in Panik geraten. |
| 7. If I did not have a data signal or could not connect to Wi-Fi, then I would constantly check to see if I had a signal or could find a Wi-Fi network | 7. Wenn ich keine mobilen Daten empfangen oder keine WLAN-Verbindung herstellen könnte, würde ich ständig prüfen, ob ich ein Signal empfangen kann oder ein WLAN-Netzwerk finde. |
| 8. If I could not use my smartphone, I would be afraid of getting stranded somewhere | 8. Wenn ich mein Smartphone nicht benutzen könnte, hätte ich Angst irgendwo zu stranden. |
| 9. If I could not check my smartphone for a while, I would feel a desire to check it | 9. Wenn ich eine Zeit lang nicht auf mein Smartphone schauen könnte, würde ich den Drang verspüren, dies zu tun. |
| If I did not have my smartphone with me, | Wenn ich mein Smartphone nicht bei mir hätte,... |
| 10. I would feel anxious because I could not instantly communicate with my family and/or friends | 10. würde ich mich ängstlich fühlen, weil ich nicht sofort mit meiner Familie und/oder Freunden kommunizieren könnte. |
| 11. I would be worried because my family and/or friends could not reach me | 11. wäre ich besorgt, weil meine Familie und/oder Freunde mich nicht erreichen könnten. |
| 12. I would feel nervous because I would not be able to receive text messages and calls | 12. wäre ich nervös, weil ich keine Textnachrichten und Anrufe empfangen könnte. |
| 13. I would be anxious because I could not keep in touch with my family and/or friends | 13. würde ich mich ängstlich fühlen, weil ich nicht mit meiner Familie und/oder Freunden in Kontakt bleiben könnte. |
| 14. I would be nervous because I could not know if someone had tried to get a hold of me | 14. wäre ich nervös, weil ich nicht wüsste, ob jemand versucht hat mich zu erreichen. |
| 15. I would feel anxious because my constant connection to my family and friends would be broken | 15. würde ich mich ängstlich fühlen, weil die permanente Verbindung zu meiner Familie und meinen Freunden unterbrochen wäre. |
| 16. I would be nervous because I would be disconnected from my online identity | 16. wäre ich nervös, weil ich von meiner Online-Identität getrennt wäre. |
| 17. I would be uncomfortable because I could not stay up-to-date with social media and online networks | 17. würde ich mich unwohl fühlen, weil ich über das, was in den sozialen Medien und Online-Netzwerken passiert, nicht auf dem Laufenden wäre. |
| 18. I would feel awkward because I could not check my notifications for updates from my connections and online networks | 18. wäre es mir unangenehm, weil ich neue Benachrichtigungen meiner Kontakte und Online-Netzwerke verpassen würde. |
| 19. I would feel anxious because I could not check my email messages | 19. würde ich mich ängstlich fühlen, weil ich meine E-Mails nicht abrufen könnte. |
| 20. I would feel weird because I would not know what to do | 20. würde ich mich komisch fühlen, weil ich nicht wüsste, was ich tun sollte. |

*Note*: answer options in English/ German: 1 = Strongly disagree/ starke Ablehnung to 7 = Strongly agree/ starke Zustimmung

The German Version of the 10 Item Big Five Inventory (BFI-10) was used to measure the five personality dimensions extraversion, neuroticism, openness to experience, conscientiousness, and agreeableness. Items are to be answered on a five-point rating scale from 1 = disagree strongly to 5 = agree strongly. For each dimension the mean of the two items was calculated. The retest reliability raged from $r$ = .66 to $r$ = .87 (interval of six weeks) [23].

At measurement time 2, the following questionnaires were used for validation:

The Depression Anxiety Stress Scale– 21 Items (DASS-21) measures depression, anxiety, and stress experienced during the last week, with seven items each (answer format: 0 = did not apply to me at all, 1 = applied to me to some degree, or some of the time, 2 = applied to me to a considerable degree or a good part of the time, and 3 = applied to me very much or most of the time). Thus, each scale had a range of 0–21 [24, 25]. Cronbach's alphas were .88 for depression, .76 for anxiety, and .86 for stress [25]. In this study, Cronbach's alpha was .82 for depression, .76 for anxiety, and .88 for stress.

The "WHO-5 Well-Being Index" (WHO-5) is a five-item, one-dimensional self-report measure of positive aspects of psychological well-being in adolescents and adults. On a 6-point Likert scale, items can be indicated as 0 (at no time), 1 (some of the time), 2 (less than half of the time), 3 (more than half of the time), 4 (most of the time) and 5 (all of the time). A sum value is formed for evaluation (range from 0 to 25). A score below 13 can indicate possible depression [26]. In a representative study, Cronbach's alpha was .92 [27]; in this study, Cronbach's alpha was .75.

The "Satisfaction with Life Scale" (SWLS) contains 5 Items with a 7-point Likert scale from 1 (strongly disagree) to 7 (strongly agree). Sum values were calculated [28, 29]. Cronbach's alpha was .78 in the original study [28], .92 in the German version [30] and .81 in this study.

## Statistical analysis

Data evaluation was carried out with the programming language R (version 4.0.5) and development interface R-Studio (version 1.4.1106) for macOS. Exploratory factor analysis (principal component analysis with varimax rotation) and confirmatory factor analyses were calculated. To determine the reliability of the NMP-Q-D, internal consistency using Cronbach's alpha, split-half reliability and retest reliability were calculated. Pearson correlation coefficients were computed to determine criterion and construct validity. Composite reliability (*CR*) and average variance extracted (*AVE*) were calculated [31]. A multiple regression analysis was used to explore the psychological predictors of nomophobia.

## Ethics

The study was approved by the Ethics Committee of the PFH University of Applied Science. Written informed consent was obtained from each participant.

## Results

The NMP-Q-D has a score range from 20 to 140. In the survey (measurement time 1), respondents scored between 22 and 127, with a mean of 62.06 and a standard deviation of 19.35. A total of 375 respondents (46.5%) scored between 21 and 59 and were within the no or mild nomophobia range. Another 399 subjects (49.4%) scored between 60 and 99, which corresponds to a moderate level of nomophobia. 33 subjects (4.1%) scored 100 or higher and thus fell into the range of severe nomophobia. Daily time spent on the smartphone ranged from 10 minutes to 697 minutes (11 hours and 37 minutes), the smartphone was used for an average of 4 hours and 16 minutes per day (*M* = 256 minutes, *SD* = 117 minutes). For smartphone

activations per day, a mean of 63.88 with a standard deviation of 47.87 could be found (range from 1 to 345 activations per day).

## Item analysis

Item descriptions are shown in Table 2. Item difficulty ranged between .16 and .62. The corrected item-total correlations reach at least medium and mostly high values, indicating that the items are strongly related to the factor to which they were assigned.

## Exploratory factor analysis

To test prerequisites for the principal component analysis, the correlations between the items were checked using a correlation matrix. There were sufficient correlations above .30, as well as no particularly high correlations above .90 that would indicate redundant items. Thus, an appropriate correlation between the items can be assumed. Furthermore, Bartlett's test for

**Table 2. Item analysis and explorative factor analysis.**

| Items | Mean | SD | Skew-ness | Kurtosis | Difficulty | Corrected item-total correlation (for each factor) | Alpha if item deleted (for each factor) | Factor 1 | Factor 2 | Factor 3 | Factor 4 | Communalities |
|---|---|---|---|---|---|---|---|---|---|---|---|---|
| Item 1 | 4.24 | 1.44 | -0.11 | -0.57 | .54 | .72 | .77 | .159 | .175 | **.769** | .071 | .652 |
| Item 2 | 4.72 | 1.52 | -0.37 | -0.57 | .62 | .69 | .74 | .149 | .070 | **.825** | .107 | .719 |
| Item 3 | 3.58 | 1.66 | 0.22 | -0.84 | .43 | .56 | .81 | .201 | .227 | **.640** | .132 | .519 |
| Item 4 | 4.53 | 1.52 | -0.24 | -0.63 | .59 | .66 | .76 | .140 | .108 | **.765** | .269 | .688 |
| Item 5 | 2.97 | 1.70 | 0.70 | -0.37 | .33 | .57 | .66 | .335 | .096 | .246 | **.688** | .656 |
| Item 6 | 2.05 | 1.35 | 1.42 | 1.55 | .18 | .58 | .67 | .149 | .298 | .162 | **.723** | .660 |
| Item 7 | 3.13 | 1.67 | 0.48 | -0.67 | .36 | .52 | .68 | .156 | .302 | .341 | **.496** | .477 |
| Item 8 | 2.87 | 1.72 | 0.74 | -0.45 | .31 | .45 | .71 | .288 | .085 | .076 | **.644** | .511 |
| Item 9 | 3.82 | 1.69 | 0.09 | -0.89 | .47 | .41 | .73 | .215 | .337 | .384 | *.219* | .355 |
| Item 10 | 3.25 | 1.70 | 0.50 | -0.64 | .38 | .78 | .91 | **.772** | .153 | .125 | .331 | .744 |
| Item 11 | 4.06 | 1.85 | -0.03 | -1.12 | .51 | .79 | .91 | **.859** | .044 | .162 | .105 | .777 |
| Item 12 | 3.20 | 1.68 | 0.48 | -0.66 | .37 | .78 | .91 | **.765** | .261 | .205 | .196 | .733 |
| Item 13 | 3.27 | 1.69 | 0.46 | -0.63 | .38 | .81 | .90 | **.805** | .194 | .145 | .245 | .767 |
| Item 14 | 3.44 | 1.78 | 0.33 | -0.97 | .41 | .73 | .92 | **.754** | .195 | .247 | .104 | .678 |
| Item 15 | 2.81 | 1.58 | 0.74 | -0.21 | .30 | .78 | .91 | **.750** | .330 | .127 | .214 | .733 |
| Item 16 | 1.54 | 0.90 | 1.91 | 3.68 | .09 | .58 | .72 | .180 | **.730** | .075 | .166 | .598 |
| Item 17 | 1.95 | 1.19 | 1.42 | 1.81 | .16 | .70 | .67 | .128 | **.815** | .206 | .095 | .732 |
| Item 18 | 2.09 | 1.25 | 1.23 | 1.20 | .18 | .63 | .69 | .244 | **.714** | .249 | .092 | .640 |
| Item 19 | 2.01 | 1.32 | 1.53 | 2.03 | .17 | .40 | .77 | .185 | **.559** | .024 | .062 | .351 |
| Item 20 | 2.53 | 1.56 | 0.84 | -0.14 | .26 | .47 | .76 | .042 | **.569** | .193 | .299 | .452 |

*Note*: varimax rotation, eigenvalue distribution: 7.926; 1,928; 1,535; 1,052; .866; .820; .688 . . .; factor loads ≥ .40 are printed in bold; factor 1: not being able to communicate, factor 2: losing connectedness, factor 3: not being able to access information, factor 4: giving up convenience

sphericity was performed, and the Kaiser-Meyer-Olkin coefficient was calculated. Bartlett's test was significant ($x^2{}_{190}$ = 8181.847, $p < .001$), therefore relationships between the items can be assumed. The Kaiser-Meyer-Olkin coefficient, with an overall value of .92 and with values of the individual items $\geq$ .84, was in the very good range and above the cut-off value of 0.5, above which a principal component analysis is appropriate.

Based on Cattell's scree test and the Kaiser-Gutman criterion, a 4-factor structure was chosen, which explained 62% of the total variance. Table 2 shows the loadings of the items on the four factors after conducting a principal component analysis with varimax rotation. A factor loading of .50 was used as the cut-off value. Overall, the items loaded predominantly on a single factor, while loadings with the other factors were very low for most items. All items except item 9 show high loading on the same factor as in the original version of the questionnaire. Likewise, the magnitude of the factor loadings is very similar to the original version. Item 9, on the other hand, shows a less clear loading on a single factor. In the original version of the NMP-Q, item 9 loads most strongly on factor 4 [7]. However, in the present German translation, the strongest loading of item 9 refers to factor 3. Item 9 does not load significantly lower on the other factors, so that an assignment to factor 4, as in the original study, also seems quite reasonable. Based on the reliability analysis presented in the following chapter, Item 9 was ultimately assigned to Factor 4. Above all, the internal consistency of the 3rd factor increases without item 9, which is why the item does not seem to make sense there. However, when item 9 is assigned to factor 4, the internal consistency of the factor increases. Due to the almost identical results with the original version, the factors can be adopted in terms of content. Overall, 62% of the variance can be explained by all four factors, with the first factor explaining 21%, the second and third factors explaining 15%, and the fourth factor explaining 11% of the variance.

## Confirmatory factor analysis

For confirmatory factor analysis, three models were tested: A unidimensional model (model 1), a correlated four-factor model (model 2), and a hierarchical model with one second-order factor explaining the four nomophobia factors (model 3). Model 1 showed a poor fit to the data (see Table 3), while model 2 and model 3 fitted well. It is thus appropriate to consider four factors and give one overall score for nomophobia. Figs 1–3 depict standardized factor loadings. Fig 2 shows the intercorrelations between the four factors.

**Table 3. Fit statistics for confirmatory factor analysis.**

|  | Model 1 | Model 2 | Model 3 |
|---|---|---|---|
| $\chi 2$ | 2533.55 | 855.46 | 859.11 |
| df | 170 | 164 | 166 |
| $\chi 2/df$; p | 14.90; $p < .001$ | 5,21; $p < .001$ | 5,17; $p < .001$ |
| CFI | 0.707 | 0.914 | 0.914 |
| TLI | 0.673 | 0.901 | 0.902 |
| RMSEA [90% CI] | 0.131 [0.127; 0.136] | 0.072 [0.068; 0.077] | 0.072 [0.066; 0.077] |
| SRMR | 0.097 | 0.053 | 0.053 |
| GFI | 0.688 | 0.902 | 0.902 |
| AGFI | 0.615 | 0.874 | 0.875 |

Note: N = 708, model 1: unidimensional model; model 2: correlated four-factor model; model 3: hierarchical model with one second-order factor explaining the four nomophobia factors

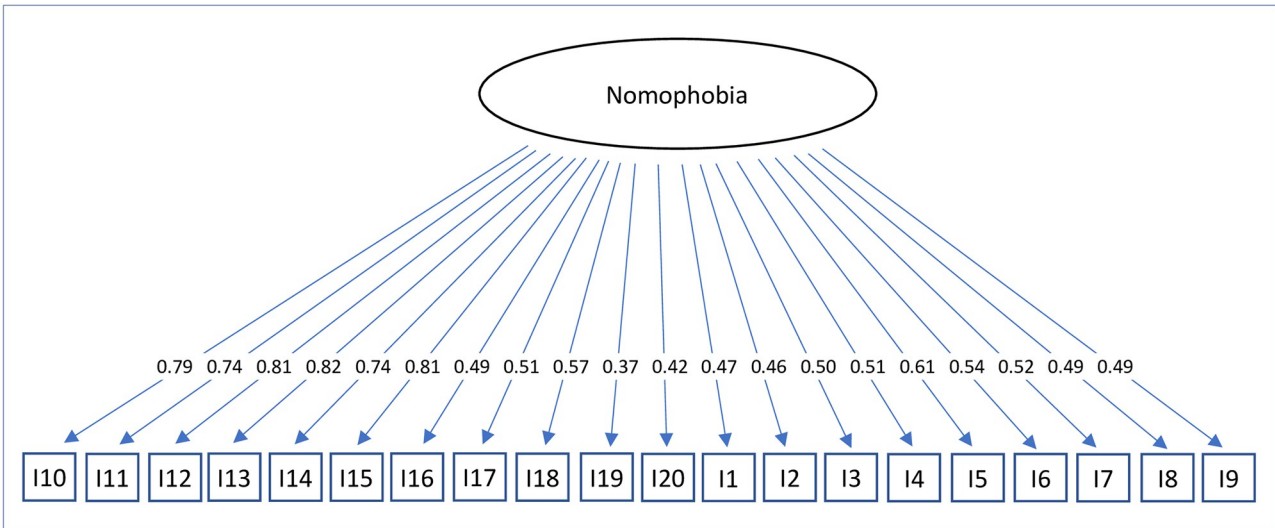

**Fig 1. Model 1: Unidimensional model of nomophobia.**

## Reliability

Cronbach's alpha for the overall NMP-Q-D was .92 (.93 at the second measurement time), split-half reliability with Spearman-Brown correction was .94 (.95 at the second measurement time), and retest reliability .80 (details see Table 4). Cronbach's alpha ranged from .74 to .92 and retest reliability from .54 to .71 for the four factors. Table 2 additionally shows the corrected item-total correlations and *"Cronbach's alpha if item deleted" values* for the respective items in relation to the associated factor. The Cronbach's alpha values show that, if the

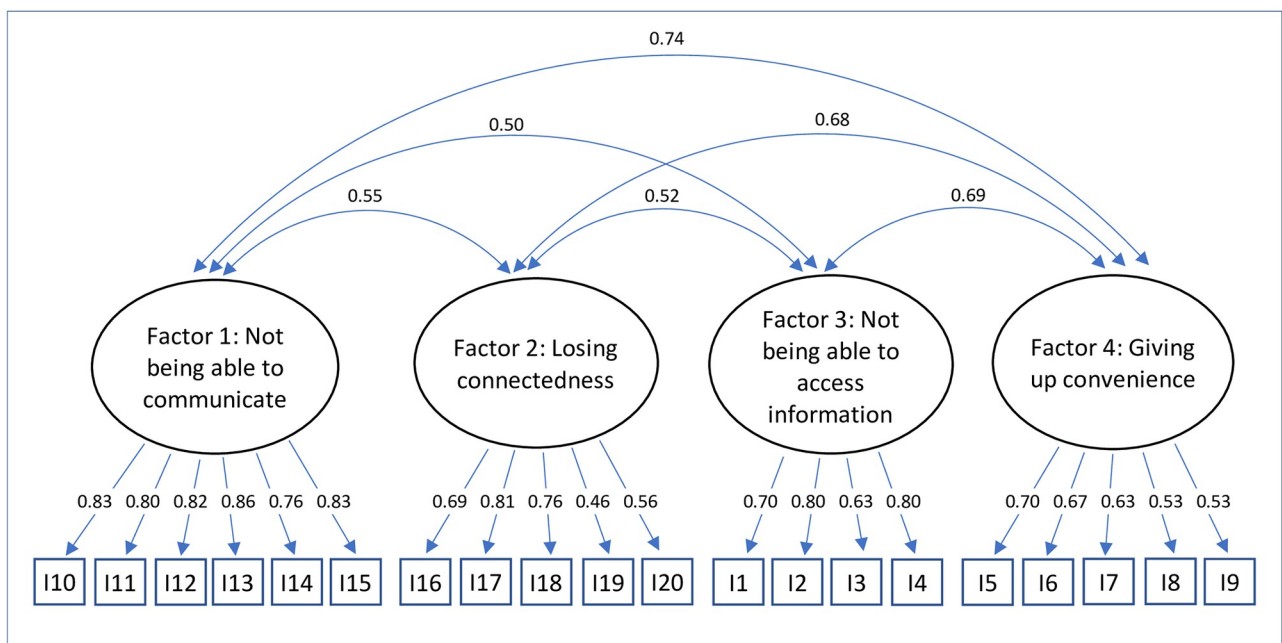

**Fig 2. Model 2: Correlated four-factor model of nomophobia.**

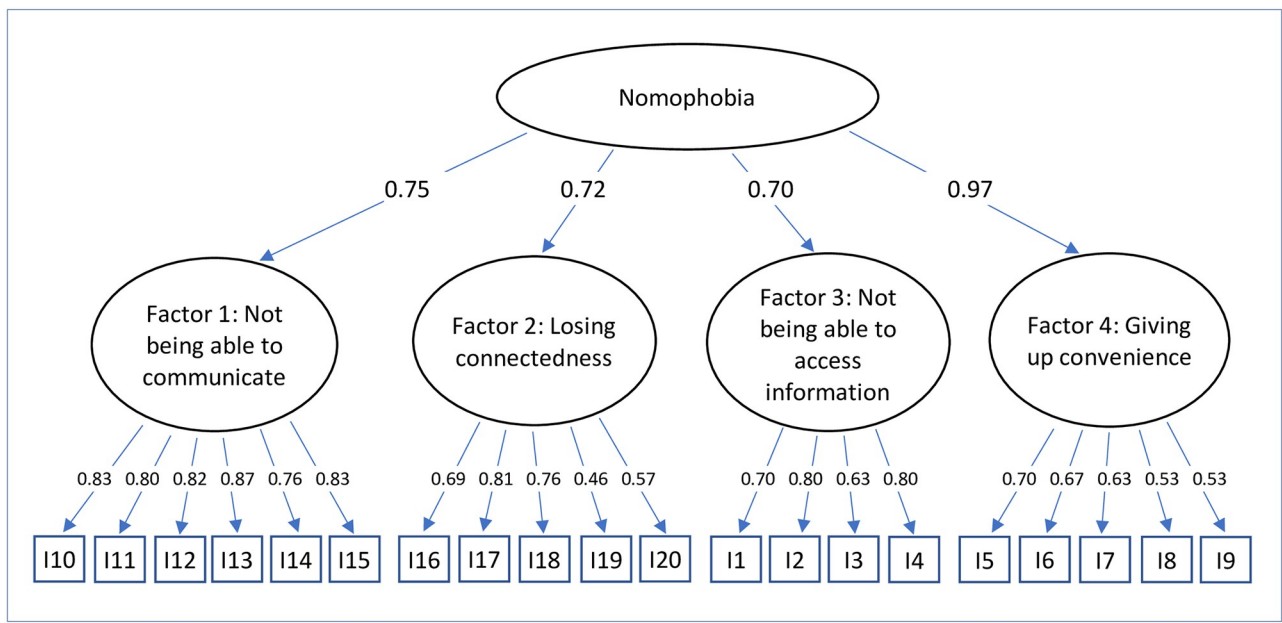

**Fig 3. Model 3: Hierarchical model with one second-order factor explaining four nomophobia factors.**

respective item were deleted, the removal of no item would lead to an increase in the internal consistency of the factor.

There were no significant mean differences between the two measurement times (see Table 5). In addition, the correlation (reliability) between the minutes of smartphone usage was .70 (p < .001) and of smartphone activation .57 (p < .001) for the two measurement times.

## Validity

NMO-Q-D correlated significantly with how long and how frequently the smartphone was used per day (see Tables 6 and 7). For the first measurement time point, r was .20 (p < .001) resp. .21 (p < .001), for the second measurement, r was .36 (p = .016) resp. .39 (p = .009), confirming the criterion validity of the NMP-Q-D. To determine convergent validity, a significant correlation of r = .38 (p < .001) was found between the NMP and the FoMO questionnaires. Also, a highly significant correlation of r = .60 (p < .001) was found between the NMP-Q-D

**Table 4. Reliability of the NMP-Q-D.**

| Factors | Cronbach's alpha (time 1) | Split-Half-Reliability (time 1) | Cronbach's alpha (time 2) | Split-Half-Reliability (time 2) | Retest-Reliability |
|---|---|---|---|---|---|
| NMP factor 1 | .92 | .86 (.92) | .92 | .82 (.90) | .68 |
| NMP factor 2 | .77 | .64 (.78) | .74 | .60 (.75) | .55 |
| NMP factor 3 | .81 | .61 (.76) | .77 | .53 (.69) | .54 |
| NMP factor 4 | .74 | .59 (.74) | .76 | .64 (.78) | .71 |
| NMP-Q-D overall | .92 | .89 (.94) | .93 | .91 (.95) | .80 |

*Note*: N measure time 1 = 708, N measure time 2 = 50. in brackets: split-half by odd-even, Split-Half-Reliability after Spearman-Brown-correction, factor 1: not being able to communicate, factor 2: losing connectedness, factor 3: not being able to access information, factor 4: giving up convenience, 5 months between measure 1 and measure 2

**Table 5. Mean differences between measurement time 1 and 2.**

|  | Time 1 | | Time 2 | | difference | T | df | p (2-tailed) |
|---|---|---|---|---|---|---|---|---|
|  | **M** | **SD** | **M** | **SD** |  |  |  |  |
| NMP factor 1 | 3.36 | 1.40 | 3.59 | 1.29 | -0.23 | -1.49 | 49 | 0.142 |
| NMP factor 2 | 2.11 | 0.89 | 2.18 | 0.93 | -0.07 | -0.55 | 49 | 0.583 |
| NMP factor 3 | 4.37 | 1.06 | 4.35 | 0.98 | 0.02 | 0.11 | 49 | 0.914 |
| NMP factor 4 | 3.17 | 1.22 | 3.25 | 1.08 | -0.08 | -0.64 | 49 | 0.527 |
| NMP-Q-D overall | 3.20 | 0.99 | 3.30 | 0.84 | -0.10 | -1.22 | 49 | 0.228 |
| Smartphone usage | 253.18 | 134.92 | 233.57 | 104.87 | 19.61 | 1.34 | 43 | 0.188 |
| Smartphone activity | 65.60 | 45.74 | 68.69 | 36.87 | -3.09 | -0.53 | 44 | 0.599 |

*Note*: N measure time 1 = 708, N measure time 2 = 50, factor 1: not being able to communicate, factor 2: losing connectedness, factor 3: not being able to access information, factor 4: giving up convenience, 5 months between measure 1 and measure 2

and the SAS-SV. No significant correlation could be shown between the NMP-Q-D and the agreeableness subscale of the BFI-10 with $r = -.01$ ($p = .702$), indicating discriminant validity. Nomophobia and neuroticism correlated positively ($r = .19$, $p < .001$), but not as strongly as neuroticism with FoMO ($r = .32$, $p < .001$) or SAS ($r = .25$, $p < .001$). A small significant negative correlation was found between nomophobia and conscientiousness ($r = -.12$, $p < .001$), openness to experience (-.08, $p = .034$) and age (-.09, $p = .008$).

At measurement time 2 (see Table 6), depression, anxiety, and stress correlated positively with nomophobia. Because of the small sample size, only the correlation to stress is significant ($r = .32$, $p = .025$). Factor 4 of NMO-Q-D ("giving up convenience") correlated significantly negatively with well-being ($r = -.29$, $p = .041$) and factor 3 ("inability to access information") correlated significantly positively with life satisfaction ($r = .30$, $p = .034$).

Composite reliability (*CR*) and average variance extracted (*AVE*) were calculated to obtain additional evidence on convergent and discriminant validity for NMP-Q and the validation scales (see Tables 6 and 7). With the exception of Agreeableness (*CR* = .675), all *CR* values were above the threshold of .70. If the *CR* is higher than .60, convergent validity can be acceptable for the factors having an *AVE* value below .50 [31]. For most correlations, the square roots of the *AVEs* were larger than the cross-correlations, suggesting discriminant validity. Lower discriminant validity was found for nomophobia factor 4 (giving up convenience), which shares a common variance with factor 1 (not being able to communicate). Low discriminant validity was also found for the DASS-21: Here, depression, anxiety, and stress were highly correlated.

## Multiple regression analysis

A multiple regression analysis was conducted to explore the psychological predictors of nomophobia at measurement time 1. The variables age, gender, duration of smartphone usage, number of smartphone activations, and the "Big 5" personality traits were included in the linear model. The results of the multiple regression analysis are shown in Table 8. According to the results, the variables time spent daily on a smartphone, number of smartphone activations, gender, and the personality dimension neuroticism had a significant influence on the nomophobia score. However, the model only explains a total of 10% of the variance in nomophobia scores.

## Gender differences

Men averaged a nomophobia score of 54.17 and women of 62.86. A significant gender difference with an effect size $d = 0.46$ was found (see Table 9); the gender effects were prominent for

**Table 6. Intercorrelation and validation at measure time 1.**

| | | M | SD | CR | AVE | N | 1 | 2 | 3 | 4 | 5 | 6 | 7 | 8 | 9 | 10 | 11 | 12 | 13 | 14 | 15 |
|---|---|---|---|---|---|---|---|---|---|---|---|---|---|---|---|---|---|---|---|---|---|
| 1 | NMP factor 1 | 20.03 | 8.74 | .923 | .668 | 807 | .817 | | | | | | | | | | | | | | |
| 2 | NMP factor 2 | 10.12 | 4.54 | .796 | .448 | 807 | .484** | .669 | | | | | | | | | | | | | |
| 3 | NMP factor 3 | 17.07 | 4.93 | .821 | .536 | 807 | .454** | .446** | .732 | | | | | | | | | | | | |
| 4 | NMP factor 4 | 14.84 | 5.69 | .750 | .379 | 807 | .622** | .568** | .564** | .615 | | | | | | | | | | | |
| 5 | Nomophobia (NMP-Q-D total) | 62.06 | 19.35 | .954 | .514 | 807 | .864** | .734** | .731** | .853** | .717 | | | | | | | | | | |
| 6 | Fear of missing out (FoMo) | 2.37 | 0.61 | .724 | .246 | 807 | .269** | .360** | .275** | .328** | .373** | .496 | | | | | | | | | |
| 7 | Smartphone-Addiction (SAS) | 26.76 | 8.32 | .839 | .349 | 807 | .391** | .568** | .478** | .568** | .599** | .501** | .591 | | | | | | | | |
| 8 | Neuroticism (BFI-10) | 3.17 | 1.02 | .827 | .705 | 806 | .179** | .153** | .087* | .178** | .191** | .321** | .247** | .839 | | | | | | | |
| 9 | Extraversion (BFI-10) | 3.04 | 1.06 | .850 | .740 | 806 | -.017 | -.017 | .066 | -.048 | -.009 | -.041 | -.026 | -.296** | .860 | | | | | | |
| 10 | Openness (BFI-10) | 3.38 | 1.02 | .822 | .698 | 806 | -.074* | -.014 | -.073* | -.065 | -.075* | .017 | -.043 | ,018 | .108** | .836 | | | | | |
| 11 | Agreeableness (BFI-10) | 3.12 | 0.82 | .675 | .518 | 806 | .014 | -.018 | -.034 | -.023 | -.013 | .026 | .027 | -.094** | .163** | .048 | .720 | | | | |
| 12 | Consciousness (BFI-10) | 3.48 | 0.82 | .774 | .631 | 806 | -.004 | -.174** | -.127** | -.164** | -.123** | -.193** | -.374** | -,090* | .112** | .003 | .020 | .794 | | | |
| 13 | Smartphone usage | 255.97 | 116.78 | | | 794 | .095* | .223** | .201** | .185** | .201** | .226** | .363** | ,144** | -.088** | -.041 | -.076* | -.292** | | | |
| 14 | Daily smartphone activations | 63.88 | 47.87 | | | 785 | .118** | .153** | .228** | .209** | .208** | .159** | .263** | 0,056 | .061 | -.021 | -.081* | -.132** | .301** | | |
| 15 | Age | 25.00 | 9.26 | | | 802 | -.109** | -.093** | .007 | -.084* | -.094** | -.341** | -.313** | -,198** | .095** | .036 | .017 | .152** | -.326** | -.198** | |
| 16 | Gender (1 = female, 2 = male) | 1.09 | 0.29 | | | 796 | -.149** | .020 | -.058 | -.152** | -.132** | -.070* | -.129* | -.258** | .087** | .068 | .052 | .012 | -.144** | .022 | .217* |

*Note*: NMP factor 1: not being able to communicate, NMP factor 2: losing connectedness, NMP factor 3: not being able to access information, NMP factor 4: giving up convenience; smartphone usage: time spent daily on a smartphone in minutes; square roots of *AVE* are shown in the diagonal (italics).

* *p* < .05,

** *p* < .01, two-tailed

**Table 7. Intercorrelation and validation at measure time 2.**

| | | M | SD | CR | AVE | N | 1 | 2 | 3 | 4 | 5 | 6 | 7 | 8 | 9 | 10 | 11 | 12 | 13 |
|---|---|---|---|---|---|---|---|---|---|---|---|---|---|---|---|---|---|---|---|
| 1 | NMP factor 1 | 21.56 | 7.71 | .899 | .602 | 50 | *.776* | | | | | | | | | | | | |
| 2 | NMP factor 2 | 1.88 | 4.65 | .788 | .440 | 50 | .426** | *.663* | | | | | | | | | | | |
| 3 | NMP factor 3 | 17.40 | 3.90 | .801 | .514 | 50 | .250 | .343* | *.717* | | | | | | | | | | |
| 4 | NMP factor 4 | 16.24 | 5.40 | .748 | .377 | 50 | .678** | .486** | .406** | *.614* | | | | | | | | | |
| 5 | Nomophobia (NMP-Q-D total) | 66.08 | 16.79 | .948 | .488 | 50 | .854** | .709** | .573** | .863** | *.698* | | | | | | | | |
| 6 | Depression (DASS-21) | 4.44 | 3.74 | .857 | .452 | 50 | .220 | .125 | -.036 | .236 | .203 | *.673* | | | | | | | |
| 7 | Anxiety (DASS-21) | 3.40 | 3.50 | .793 | .350 | 50 | .298* | .117 | -.007 | .120 | .206 | .596** | *.592* | | | | | | |
| 8 | Stress (DASS-21) | 6.40 | 4.76 | .895 | .542 | 50 | .377** | -.001 | .080 | .388** | .317* | .617** | .583** | *.736* | | | | | |
| 9 | Wellbeing (WHO5) | 12.42 | 4.12 | .755 | .388 | 50 | -.148 | -.144 | -.006 | -.291* | -.203 | -.676** | -.394** | -.608** | *.623* | | | | |
| 10 | Life satisfaction (SWLS) | 25.14 | 4.50 | .823 | .488 | 50 | -.057 | .054 | .301* | -.001 | .058 | -.460** | -.320* | -.364** | .532** | *.699* | | | |
| 11 | Smartphone usage | 233.57 | 104.87 | | | 44 | .226 | .258 | .303* | .353** | .360** | .112 | .128 | -.031 | -.253 | -.022 | | | |
| 12 | Daily smartphone activations | 68.69 | 36.87 | | | 45 | .142 | .283 | .308* | .527** | .388** | .093 | -.062 | .231 | -.296* | -.068 | .416** | | |
| 13 | Age | 24.10 | 6.12 | | | 50 | .139 | -.108 | .173 | .079 | .100 | .025 | -.004 | .040 | .022 | .029 | -.029 | -.023 | |
| 14 | Gender (1 = female, 2 = male) | 1.10 | 0.30 | | | 50 | -.173 | .038 | .138 | -.115 | -.074 | -.220 | -.096 | -241 | .194 | .318* | .264 | .019 | -.050 |

*Note*: NMP factor 1: not being able to communicate, NMP factor 2: losing connectedness, NMP factor 3: not being able to access information, NMP factor 4: giving up convenience; smartphone usage: time spent daily on a smartphone in minutes; square roots of *AVE* are shown in the diagonal (italics).

* $p < .05$,

** $p < .01$, two-tailed

**Table 8. Multiple regression analysis at measure time 1.**

| Items | B | Standard error | Beta | T | p value |
|---|---|---|---|---|---|
| **Overall** | | | | | |
| (Constant) | 49.61 | 7.09 | – | 7.00 | < .001 |
| Smartphone usage | 0.02 | 0.01 | 0.13 | 3.29 | .001 |
| Smartphone activations | 0.06 | 0.02 | 0.15 | 3.96 | < .001 |
| Gender | -5.85 | 2.43 | -0.09 | -2.42 | .016 |
| Age | 0.08 | 0.08 | 0.04 | 1.01 | .314 |
| Neuroticism | 3.10 | 0.71 | 0.16 | 4.37 | < .001 |
| Extraversion | 1.31 | 0.68 | 0.07 | 1.94 | .053 |
| Openness | -1.10 | 0.66 | -0.06 | -1.66 | .096 |
| Agreeableness | 0.57 | 0.84 | 0.02 | 0.68 | .497 |
| Conscientiousness | -1.26 | 0.86 | -0.05 | -1.46 | .146 |

*Note*: N = 766; R = .32, gender: 1 = female, 2 = male, smartphone usage: time spent daily on a smartphone; smartphone activations: daily activations of the smartphone

**Table 9. Gender differences.**

| | female | | | male | | | differences | | df | p value (2- tailed) | d |
|---|---|---|---|---|---|---|---|---|---|---|---|
| | N | M | SD | N | M | SD | M | T | | | |
| Factor_1 | 721 | 20.47 | 8.70 | 75 | 16.01 | 7.99 | 4.46 | 4.25 | 794 | < .001 | 0.53 |
| Factor_2 | 721 | 10.13 | 4.52 | 75 | 9.83 | 4.67 | 0.30 | 0.55 | 794 | .583 | 0.07 |
| Factor_3 | 721 | 17.16 | 4.87 | 75 | 16.17 | 5.40 | 0.98 | 1.64 | 794 | .101 | 0.19 |
| Factor_4 | 721 | 15.11 | 5.64 | 75 | 12.16 | 5.28 | 2.95 | 4.34 | 794 | < .001 | 0.54 |
| NMP-Q-D | 721 | 62.86 | 19.15 | 75 | 54.17 | 18.96 | 8.69 | 3.74 | 794 | < .001 | 0.46 |
| FoMO | 721 | 2.38 | 0.60 | 75 | 2.23 | 0.70 | 0.15 | 1.98 | 794 | .049 | 0.23 |
| SAS | 721 | 27.01 | 8.00 | 75 | 23.38 | 9.41 | 3.63 | 3.68 | 794 | < .001 | 0.42 |
| Neuroticism | 720 | 3.25 | 0.99 | 75 | 2.35 | 0.95 | 0.90 | 7.53 | 793 | < .001 | 0.93 |
| Extraversion | 720 | 3.01 | 1.06 | 75 | 3.33 | 0.97 | -0.32 | -2.47 | 793 | .014 | 0.31 |
| Openness | 720 | 3.36 | 1.02 | 75 | 3.59 | 0.93 | -0.24 | -1.91 | 793 | .056 | 0.24 |
| Agreeableness | 720 | 3.11 | 0.81 | 75 | 3.25 | 0.85 | -0.15 | -1.47 | 793 | .141 | 0.18 |
| Consciousness | 720 | 3.49 | 0.80 | 75 | 3.52 | 0.90 | -0.03 | -0.34 | 793 | .736 | 0.04 |
| Smartphone usage | 711 | 260.76 | 113.72 | 73 | 203.14 | 128.13 | 57.62 | 4.07 | 782 | < .001 | 0.48 |
| Smartphone activ. | 701 | 64.22 | 47.36 | 73 | 60.58 | 51.61 | 3.65 | 0.62 | 772 | .535 | 0.07 |

*Note*: factor 1: not being able to communicate, factor 2: losing connectedness, factor 3: not being able to access information, factor 4: giving up convenience; FoMO: fear of missing out; SAS: Smartphone-Addiction(-Scale), smartphone usage: time spent daily on a smartphone, smartphone activ.: daily activations of the smartphone, smartphone usage: time spent daily on a smartphone, smartphone activ.: daily activations of the smartphone

factor 1 ("not being able to communicate", $d$ = 0.53) and factor 4 ("giving up convenience", $d$ = 0.54). For factors 2 ("losing connectedness") and 3 ("not being able to access information"), there were no significant gender differences. Smartphone addiction was also significantly higher in women than in men ($d$ = 0.42); the same applies to fear of missing out, whereby the effect was smaller ($d$ = 0.23). Significant gender differences were found for two of the Big Five, so women scored higher in neuroticism ($d$ = 0.93) and lower in extraversion ($d$ = 0.31). Women were significantly longer on the smartphone than men ($d$ = 0.48), but there was no significant difference in activations per day.

## Discussion

The results showed that many subjects had medium to high values, suggesting that nomophobia is also widespread in Germany. A 4-factor structure of the NMP-Q could also be confirmed in the German version. Based on the reliability analysis, a good internal consistency with $\alpha$ = .92 of the NMP-Q-D was confirmed, which is comparable to the internal consistency of the original version of the questionnaire ($\alpha$ = .95) [7]. The individual factors of the NMP-Q-D achieved reliability values between $\alpha$ = .77 and $\alpha$ = .92, which can be rated as acceptable to very good. The retest reliability for the overall-NMP-Q-D score of .80 is good; for the four factors of nomophobia, the correlation between the two survey times ranged from .54 to .71. During a 5-month period the components of nomophobia can change, which can be a positive aspect for the therapy of nomophobia. The item difficulties are mostly in the medium range, which allows for a good differentiation.

Significant positive correlations with duration of smartphone usage per day as well as the number of smartphone activations per day confirmed criterion validity of the NMP-Q-D. The construct validity can also be confirmed based on the results of the convergent and discriminant validity. As expected, a significant overlap between nomophobia and Fear of Missing Out

(FoMO) could be assumed. Additionally, nomophobia is often associated with smartphone addiction. The middle-ranged correlations with FoMO may be explained by the relatively low *AVE* of the FoMO scale. The fact that even small correlations (e.g. NMP-Q-D and age correlated with $r = -.094$) became significant in sample 1 is due to the relatively large sample size.

In the retest-study, the sample was smaller. Therefore, a correlation coefficient of $\geq .29$ can be considered significant here. The descriptive correlation coefficients reported here are comparable to other studies but might not be significant only because of the smaller sample size. In this study, the correlation with depression was .20; in the two other studies both correlations were .23 [5, 16]. For anxiety, it was nearly the same: the correlations were .21 in this study and .21 [5] and .27 [16] in the two other studies. Finally, for stress, the correlation of .31 was significant and slightly higher than .28 reported in one other study [16].

This result of the multiple regression confirmed the influence of gender on nomophobia score found in other studies [16, 17]. Therefore, it can be assumed that, as already suspected [17], women use the smartphone more for communication due to a stronger need for social relationships and thus achieve higher nomophobia scores. The gender differences were based on the dimensions of "not being able to communicate" and "giving up convenience". Regarding smartphone usage, there were no significant differences in activation, but women were longer on the smartphone than men.

The question whether nomophobia should be included in the International Classification of Diseases (ICD) or the Diagnostic and Statistical Manual of Mental Disorders (DSM) is valid. Diagnostic criteria for nomophobia and differential diagnosis have already been recommended [32, 33]. A reasonable threshold must be defined in order to justify therapeutic intervention. In severe cases, cognitive-behavioral therapy, mindfulness, and emotion-oriented therapy can indeed help reduce symptoms [34].

Several possible limitations need to be considered for the present study. As with any translation, language-specific nuances are inevitable. Furthermore, the present convenience sample is not representative of the entire German population. Despite the large age range of 18 to 87 years, most of the test persons were between 20 and 26 years old. The gender distribution of the sample is also not representative, as only 75 of the total 807 test persons stated that they were male. Also, regarding the level of education, the present sample does not cover all groups equally. A total of 583 of the 807 respondents stated a high school leaving qualification. For this reason, the present sample is primarily representative of young women with a high level of education. Also, the generalizability of the retest study is limited due to the number of participants of 50 people. Moreover, as with any other self-reported questionnaire, the self-reported structure of the questionnaires used may be a limitation because of methodological bias, like social desirability. Since the present study is a cross-sectional study and several scales were used simultaneously, a common method bias may arise. Therefore, Harman's single factor score was calculated. A common method bias is indicated if the total variance extracted by one factor exceeds 50% [35, 36]. This, however, does not apply to this study, since this number is far lower, namely 22.64% at the first measurement time and 23.82% at the second.

## Supporting information

**S1 Data.**
(XLSX)

**S2 Data.**
(XLSX)

## Author Contributions

**Conceptualization:** Melina Coenen, Yvonne Görlich.

**Data curation:** Melina Coenen, Yvonne Görlich.

**Formal analysis:** Melina Coenen, Yvonne Görlich.

**Investigation:** Melina Coenen, Yvonne Görlich.

**Methodology:** Melina Coenen, Yvonne Görlich.

**Project administration:** Yvonne Görlich.

**Supervision:** Yvonne Görlich.

**Validation:** Yvonne Görlich.

**Writing – original draft:** Melina Coenen, Yvonne Görlich.

**Writing – review & editing:** Melina Coenen, Yvonne Görlich.

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
