## [Decision Letter · Decision Letter 0]

29 Aug 2022

PONE-D-22-09552Exploring Nomophobia with a German adaption of the Nomophobia Questionnaire (NMP-Q-D)PLOS ONE

Dear Dr. Görlich,

Thank you for submitting your manuscript to PLOS ONE. After careful consideration, we feel that it has merit but does not fully meet PLOS ONE’s publication criteria as it currently stands. Therefore, we invite you to submit a revised version of the manuscript that addresses the points raised during the review process.

The reviewer report can be found at the end of this email. Several concerns were raised, particularly with regard to the methodology and statistical analysis. Please revise your manuscript carefully to address all comments.

Please note that we have only been able to secure a single reviewer to assess your manuscript. We are issuing a decision on your manuscript at this point to prevent further delays in the evaluation of your manuscript. Please be aware that the editor who handles your revised manuscript might find it necessary to invite additional reviewers to assess this work once the revised manuscript is submitted. However, we will aim to proceed on the basis of this single review if possible. 

We look forward to receiving your revised manuscript.

Kind regards,

Debora Walker

Editorial Office

PLOS ONE

Journal Requirements:

Reviewers' comments:

Reviewer's Responses to Questions

**Comments to the Author**

1. Is the manuscript technically sound, and do the data support the conclusions?

Reviewer #1: Yes

2. Has the statistical analysis been performed appropriately and rigorously? 

Reviewer #1: Yes

3. Have the authors made all data underlying the findings in their manuscript fully available?

Reviewer #1: No

4. Is the manuscript presented in an intelligible fashion and written in standard English?

Reviewer #1: Yes

5. Review Comments to the Author

Reviewer #1: Manuscript Number: PONE-D-22-09552

Article Type: Research Article

Full Title: Exploring Nomophobia with a German adaption of the Nomophobia Questionnaire

(NMP-Q-D)

Objective: The present study aimed to translate the nomophobia questionnaire (NMP-Q) into German, validate NMP-Q-D, and use it for exploring nomophobia in relation to fear of missing out, smartphone addiction, depression, anxiety, stress, psychological well-being, satisfaction with life, and big five personality traits (i.e., extraversion, neuroticism, openness to experience, conscientiousness, and agreeableness).

The manuscript needs more work and effort in each section:

1. Since this is a cross-sectional study and applied several scales at the same time common method bias may arise from certain tendencies that respondents apply. Therefore, common method bias should be an important concern and assessed.

2. The instrument is formed by diverse demographic variables along with seven different scales including NMP-Q-D, FoMO, SAS-SV, BFI-10, DASS-21, WHO-5, and SWLS. Since this was a scale adaptation study:

a. Convergent and discriminant validity should be tested.

b. CR and AVE values should also be reported in the matrix to test convergent and discriminant validity.

c. Square root of the AVE (show in the diagonals) and AVE values should also be reported in the correlation matrix table.

d. Model fit estimates including GFI, AGFI, and SRMR should be reported.

3. Reliability and validity results of the used scales (i.e., FoMO, SAS-SV, BFI-10, DASS-21, WHO-5, and SWLS) should be reported.

4. For confirmatory factor analysis tree models were tested: A unidimensional model (model 1), an uncorrelated four-factor model (model 2) and a hierarchical model with one second-order factor explaining the four nomophobia factors (model 3). Provide a figure to visualize the models, at least the model with the best model fit. Figures should be in high resolution (TIFF images with minimum 300 dpi).

5. Table 7 shows the results of the multiple regression analysis at measure time 1. In the table, Std. Deviation can be Std. Error, please check.

6. PLOS authors have the option to publish the peer review history of their article (what does this mean?). If published, this will include your full peer review and any attached files.

Reviewer #1: **Yes: **Ibrahim ARPACI

---

## [Author Response · Author response to Decision Letter 0]

7 Oct 2022

Dear Prof. Arpaci,

Thank you very much for the evaluation of our manuscript and the valuable input for improvements. We have prepared a revised version accordingly. A list of changes and answers to all points raised follow on the next pages. The changes in the manuscript are highlighted in yellow.

Best regards

Yvonne Görlich

Changes to the manuscript:

• A new table 3 was added with the fit statistics for the confirmatory factor analysis. Three new figures illustrate the details. This is referred to in the text. 

• In tables 6 and 7, CR, AVE and square roots of AVE were added for all scales. This is referred to in the text.

• In the discussion, the point was added that this was a cross-sectional study and that a possible common method bias was checked factor-analytically via Harman's single factor score. 

 

Answers to the Reviewers’ queries

Reviewer #1: Manuscript Number: PONE-D-22-09552

Comments to the Author

1. Is the manuscript technically sound, and do the data support the conclusions?

Reviewer #1: Yes

2. Has the statistical analysis been performed appropriately and rigorously? 

Reviewer #1: Yes

3. Have the authors made all data underlying the findings in their manuscript fully available?

Reviewer #1: No

Data are now available in the supplements.

4. Is the manuscript presented in an intelligible fashion and written in standard English?

Reviewer #1: Yes

5. Review Comments to the Author

Reviewer #1: Manuscript Number: PONE-D-22-09552

Article Type: Research Article

Full Title: Exploring Nomophobia with a German adaption of the Nomophobia Questionnaire

(NMP-Q-D)

Objective: The present study aimed to translate the nomophobia questionnaire (NMP-Q) into German, validate NMP-Q-D, and use it for exploring nomophobia in relation to fear of missing out, smartphone addiction, depression, anxiety, stress, psychological well-being, satisfaction with life, and big five personality traits (i.e., extraversion, neuroticism, openness to experience, conscientiousness, and agreeableness).

The manuscript needs more work and effort in each section:

1. Since this is a cross-sectional study and applied several scales at the same time common method bias may arise from certain tendencies that respondents apply. Therefore, common method bias should be an important concern and assessed.

In the discussion, the point was made that this was a cross-sectional study and a possible common method bias was checked factor-analytically via Harman's single factor score.

2. The instrument is formed by diverse demographic variables along with seven different scales including NMP-Q-D, FoMO, SAS-SV, BFI-10, DASS-21, WHO-5, and SWLS. Since this was a scale adaptation study:

a. Convergent and discriminant validity should be tested.

Was checked via the proposed indices CR, AVE and square roots of AVE.

b. CR and AVE values should also be reported in the matrix to test convergent and discriminant validity.

In tables 6 and 7, CR and AVE were added for all scales.

c. Square root of the AVE (show in the diagonals) and AVE values should also be reported in the correlation matrix table.

In tables 6 and 7, the square roots of AVE are now shown in the diagonals for all scales.

d. Model fit estimates including GFI, AGFI, and SRMR should be reported.

A new table 3 was added, showing the fit statistics for the confirmatory factor analysis (including GFI, AGFI, and SRMR).

3. Reliability and validity results of the used scales (i.e., FoMO, SAS-SV, BFI-10, DASS-21, WHO-5, and SWLS) should be reported.

In tables 6 and 7, CR, AVE and the square roots of the AVE are also given for all scales used. This shows the reliability and validity of the scales.

4. For confirmatory factor analysis tree models were tested: A unidimensional model (model 1), an uncorrelated four-factor model (model 2) and a hierarchical model with one second-order factor explaining the four nomophobia factors (model 3). Provide a figure to visualize the models, at least the model with the best model fit. Figures should be in high resolution (TIFF images with minimum 300 dpi).

For each tested model, a figure is now provided (figure 1 to figure 3).

5. Table 7 shows the results of the multiple regression analysis at measure time 1. In the table, Std. Deviation can be Std. Error, please check.

Thank you for pointing this out, it should indeed read “Std. Error” and has been changed accordingly.

---

## [Editor Report · Decision Letter 1]

18 Oct 2022

PONE-D-22-09552R1Exploring Nomophobia with a German adaption of the Nomophobia Questionnaire (NMP-Q-D)PLOS ONE

Dear Dr. Görlich,

Thank you for submitting your manuscript to PLOS ONE. After careful consideration, we feel that it has merit but does not fully meet PLOS ONE’s publication criteria as it currently stands. Therefore, we invite you to submit a revised version of the manuscript that addresses the points raised during the review process.

We look forward to receiving your revised manuscript.

Kind regards,

Assoc. Prof. Ibrahim ARPACI

Guest Editor

PLOS ONE

Journal Requirements:

Additional Editor Comments (if provided):

There are still some minor issues that need to be fixed. I kindly ask the authors to proofread the manuscript. There are (much too) many instances of using the typos, wrong tense and voice, throughout the manuscript. For example, in Table 2 "Kommunalitäten" will be "Communalities." In Table 6, "two-sided" will be “two-tailed." Again, in Table 6, significance level of the correlation coefficients should be re-checked. For example, the correlation coefficient (r) between Extraversion and Age is reported as r= .095 **p < .01. But correlation coefficient (r) should be over 10 for a significance level of 90% (r > .10, p < .01).
---

## [Author Response · Author response to Decision Letter 1]

29 Nov 2022

Journal Requirements:

 We reviewed the reference list and found no issues.

Additional Editor Comments (if provided):

There are still some minor issues that need to be fixed. I kindly ask the authors to proofread the manuscript. There are (much too) many instances of using the typos, wrong tense and voice, throughout the manuscript. For example, in Table 2 "Kommunalitäten" will be "Communalities." In Table 6, "two-sided" will be "two-tailed." Again, in Table 6, significance level of the correlation coefficients should be re-checked. For example, the correlation coefficient (r) between Extraversion and Age is reported as r= .095 **p < .01. But correlation coefficient (r) should be over 10 for a significance level of 90% (r >.10, p < .01).

We proofread the manuscript, corrected multiple issues (e.g. tense, voice, communalities, two-tailed), and marked all corrections. 

We re-calculated the correlation scores and checked the significance levels and found no discrepancies. We added the following sentence in the discussion: “The fact that even small correlations (e.g. NMP-Q-D and age correlated with r=-.094) became significant in sample 1 is due to the relatively large sample size.”

---

## [Editor Report · Decision Letter 2]

6 Dec 2022

Exploring Nomophobia with a German adaption of the Nomophobia Questionnaire (NMP-Q-D)

PONE-D-22-09552R2

Dear Dr. Görlich,

We’re pleased to inform you that your manuscript has been judged scientifically suitable for publication and will be formally accepted for publication once it meets all outstanding technical requirements.

Kind regards,

Prof. Ibrahim Arpaci

Editor

PLOS ONE

---

## [Editor Report · Acceptance letter]

21 Dec 2022

PONE-D-22-09552R2 

Exploring Nomophobia with a German adaption of the Nomophobia Questionnaire (NMP-Q-D) 

Dear Dr. Görlich:

I'm pleased to inform you that your manuscript has been deemed suitable for publication in PLOS ONE. Congratulations! Your manuscript is now with our production department. 

Kind regards, 

on behalf of

Prof. Ibrahim Arpaci 

Guest Editor

PLOS ONE